# Replication and Spread of Oncolytic Herpes Simplex Virus in Solid Tumors

**DOI:** 10.3390/v14010118

**Published:** 2022-01-10

**Authors:** Bangxing Hong, Upasana Sahu, Matthew P. Mullarkey, Balveen Kaur

**Affiliations:** Department of Neurosurgery, McGovern Medical School, The University of Texas Health Science Center at Houston, Houston, TX 77030, USA; upasana.sahu@uth.tmc.edu (U.S.); matthew.p.mullarkey@uth.tmc.edu (M.P.M.)

**Keywords:** alphaherpesviruses, oncolytic virus, herpes simplex virus, replication

## Abstract

Oncolytic herpes simplex virus (oHSV) is a highly promising treatment for solid tumors. Intense research and development efforts have led to first-in-class approval for an oHSV for melanoma, but barriers to this promising therapy still exist that limit efficacy. The process of infection, replication and transmission of oHSV in solid tumors is key to obtaining a good lytic destruction of infected cancer cells to kill tumor cells and release tumor antigens that can prime anti-tumor efficacy. Intracellular tumor cell signaling and tumor stromal cells present multiple barriers that resist oHSV activity. Here, we provide a review focused on oncolytic HSV and the essential viral genes that allow for virus replication and spread in order to gain insight into how manipulation of these pathways can be exploited to potentiate oHSV infection and replication among tumor cells.

## 1. Introduction

Oncolytic viruses are viruses that are engineered or naturally selected to target tumor cells for infection, replication and lysis. Apart from direct tumor destruction, these viruses also stimulate host systemic anti-tumor immunity via the release of tumor-associated antigens (TAAs) and pattern recognition receptors (PRR). Numerous oncolytic viruses are currently in clinical trials, which include herpes simplex virus (HSV), adenovirus, polio and vaccinia virus; however, herpes simplex virus type 1 (HSV-1) is currently the most clinically advanced platform.

HSV-1 is a large double-stranded DNA containing enveloped virions, which has an ability to remain latent in host cell neurons [1]. It is widely prevalent, with an estimated 58% of the U.S. population and about 67% of the global population thought to have been exposed to it in their lifetime [2,3]. The mammalian herpesviruses are members of the Herpesviridae family, which can be categorized into three sub-families, including the alphaherpesvirinae, betaherpesvirinae, and gammaherpesvirinae. Viruses of the alphaherpesviruses sub-family are neuro-invasive [4] and are able to infect both the central and peripheral nervous system, allowing for viral spread within the nervous system through retrograde or antegrade transport of virions. The human alphaherpesviruses include HSV-1, HSV-2, and varicella zoster virus (VZV).

Oncolytic HSV-1-derived viruses that have been approved for treatment include T-VEC (Imlygic) and G47delta. Imlygic is deleted for viral ICP34.5 and ICP47. It also expresses human granulocyte–macrophage colony-stimulating factor (GM-CSF) and has demonstrated clinical benefit in melanoma patients [5]. Recently, the combination of T-VEC with immune checkpoint inhibitors [6,7,8] has shown enhanced efficacy in various types of tumors. Readers are referred to a recent review on this subject that details the nuances of combining oncolytic viral therapy with immune checkpoint blockade [9]. More recently, Daichii announced the conditional approval for G47Δ, another oHSV for the treatment of recurrent GBM patients in Japan. Currently, innovative research from multiple laboratories has generated numerous new-generation oHSVs that are in different stages of development.

In this review, we summarize the factors that control the infection, replication, and spread of oncolytic HSV and highlight strategies utilized to facilitate the efficacy of this approach (Table 1).

## 2. Infection and Spread of HSV-1

The process of HSV infection begins once viral glycoproteins gC [10,11] and gB [12] and glycoprotein gD on HSV-1 bind to the host cell’s viral entry receptors, such as herpesvirus entry mediator (HVEM) and Nectin-1 [13,14]. The viral glycoproteins gB, gH and gL assist in fusion, resulting in the release of virus particles into the cytoplasm [15,16,17]. The HSV-1 genome is then translocated into the cell’s nucleus [18] where it circularizes and begins replication. After replication, the genome is re-packaged and the resulting nucleocapsid fuses with the inner nuclear membrane. The de-enveloped nucleocapsid then enters the cytoplasm where it matures, obtains tegument proteins, and forms vesicles that then form the new viral envelope [19,20].

**Table 1 viruses-14-00118-t001:** Summary of cellular and viral genes described in this review that impact oHSV replication and spread.

Factor Types	Factors	Function	References
Cancer cell intrinsic factors	PKR	Block cellular protein translation	[21,22,23,24,25,26,27,28,29,30,31]
	cGAS-STING	Induce ISGs to limit virus infection	[32,33,34,35]
	Type I interferon	Activation of JAK–STAT1 and transcription of ISGs to inhibit virus infection	[36,37]
	TLR	Sense viral infection and activate NF-κB	[38,39]
Virus genes	UL23	Encodes thymidine kinase; Viral enzymes involved in nucleotide metabolism	[40]
	RL1	Virulence for neuron infection and latency	[41,42,43,44,45,46,47,48,49,50,51,52,53]
	ICP4	Immediate early (IE) protein that is required for early and late gene transcription	[54,55,56,57,58,59]
	ICP6	Encodes ribonucleotide reductase; converts ribonucleotides to deoxy-ribonucleotides; essential for HSV viral DNA replication	[60,61,62,63]
	ICP47	Viral protein downregulating antigen presentation activity by inhibiting transporter associated with antigen processing protein (TAP) function	[63]

## 3. Cancer Cell Intrinsic Barriers That Limit oHSV-1

Intracellular innate defense anti-viral defense responses play a decisive role in controlling virus-mediated tumor cell lytic death and also transgene expression. Here, we will detail some of the cell intrinsic factors that are known to play a significant role in viral efficacy (Figure 1).

PKR signaling: Protein Kinase R (PKR) signaling is the front line defense of host cells to detect and control virus infection and replication. PKR signaling is activated by dsRNA from cellular, viral or other origins (such as poly I:C), and also by growth factors, cytokines, oxidative stress or pro-inflammatory stimuli [21,22]. After virus infection, PKR signaling is activated through auto-phosphorylation and leads to the phosphorylation of eIF-2α [23,24], which blocks cellular translation and hence constitutes the major mechanism by which PKR exerts its anti-viral function [25,26]. PKR activation also activates signaling through MAPK and NF-kB signaling pathways to augment the induction of IFN and inflammatory cytokines. HSV-1 has evolved to encode for genes that can encode for proteins that disable anti-viral activation in tumor cells. HSV-1 encodes for several genes that can combat host anti-viral effects. These include viral genes such as ICP34.5, US11 and VHS. Infected cell protein 34.5 (ICP34.5), the neuro-virulence protein of HSV-1, blocks PKR activation and promotes viral replication [27]. ICP34.5 is a virulence factor contributing to increased virus replication. It also regulates cellular protein phosphatase 1 (PP1) and promotes PP1-mediated de-phosphorylation of EIF2α [28] thus allowing for the translation of host and viral mRNA to proceed. Furthermore, US11, which is expressed by HSV-1 can also be exploited to inhibit tumor cell PKR signaling after oHSV infection [29]. US11 is able to directly interact with PKR and inhibits PKR auto-phosphorylation [30], while VHS, an HSV tegument nuclease, results in cellular RNA degradation [31].

cGAS-STING signaling: dsDNA of HSV-1 in infected cells’ cytosol and endosomes can be sensed by cell cyclic GMP–AMP synthase(cGAS). Activated cGAS then synthesizes 2′,3′-cGAMP, which then activates its downstream signaling effector, the endoplasmic reticulum (ER)-localized protein, stimulator of interferon (IFN) genes (STING). STING then induces downstream autophagosome formation and the release of chemokines and cytokines to exert a potent anti-viral immune response and the upregulation of interferon-stimulated genes (ISGs) [32,33,34,35].

Type-I IFN signaling: HSV-1 infection induces type I interferon (IFNs) secretion from infected cells or immune cells. Type I IFNs bind to the cellular receptor subunits IFNα receptors (IFNAR) 1 and 2 leading to the dimerization and auto-phosphorylation of Janus kinase 1 (JAK1), which phosphorylates and activates signal transducers and activators of transcription 1 (STAT1) and leads to the transcription of ISGs that modulate the antiviral immune response [36,37].

TLR signaling: Upon viral entry, the endosomes of infected cells can detect viral nucleic acids through Toll-like receptors (TLRs), including TLR3, TLR7/8 and TLR9. The activation of TLRs in neurons and glial cells leads to the generation of cytokines that leads to an anti-viral response by recruiting macrophages and inducing the production of proteins that cause the degradation of mRNA and inhibition of translation [38]. Binding of TLRs to viral nucleic acid leads to the recruitment of the adaptor protein, MyD88, downstream IRAK and TRAF6 complex formation and NK-κB activation [39].

## 4. HSV-1 Encoded Genes That Are Manipulated to Generate Tumor Specific Oncolytic HSV

oHSVs typically are designed to contain loss of function mutations in one or more virulence genes due to safety concerns. These loss of function mutations lead to decreased oHSV infectivity, replication, and spread in tumor cells, but this defect is often complemented in malignant cells allowing for tumor-specific viral spread. In the following section we summarize current strategies used to reduce virulence while increasing viral infection and replication in tumor cells.

### 4.1. UL23

The first generation of engineered oHSV was dlsptk, a virus that was created by the deletion of UL23 in the HSV genome [40]. The UL23 gene encodes for viral thymidine kinase (TK), which is involved in the nucleotide metabolism needed for viral DNA replication. Proliferating tumor cells usually have elevated TK activity, which allows dlsptk (TK-deleted HSV-1) to infect and replicate selectively in tumor cells, sparing the non-dividing normal cells. However, since HSV TK is the target of common anti-herpetic strategies, the deletion of this gene makes the virus resistant to anti-herpetic agents frequently utilized in the clinic. Thus, while dlsptk was a very innovative proof of concept, its clinical translation has not been actively pursued [40].

### 4.2. RL1

Viral RL1 encodes for ICP 34.5 (γ34.5), which has been shown to play a significant role in HSV-1-mediated neuro-virulence. Viruses that lack a functional ICP34.5 are much safer and its deletion significantly reduces HSV-1 infection and replication in the central nervous system [41]. The γ34.5 gene also plays an important role in HSV’s infection and latency in neurons. It controls HSV’s replication by blocking the cellular anti-viral defense response-mediated phosphorylation of eIF2α and the ensuing shut off of host translation [27]. ICP34.5 also binds to cellular Beclin-1 to block autophagy and this function is linked to the viruses’ neuro-virulence. Thus, its deletion significantly attenuates the replication capacity of the virus in normal cells. Malignant cells which usually harbor a defective anti-viral IFN response thus compensate for this defect, allowing an ICP34.5-deleted virus to replicate in a cancer-cell-specific manner in neurons [42]. Thus, ICP 34.5 is frequently disabled in most of the oHSVs, being evaluated in patients.

Since gamma34.5 is essential for robust herpes simplex virus’ replication, oHSV with deletion of gamma34.5 results in a highly attenuated and safe virus at the expense of viral replication. Engineering gamma34.5 expression under a tumor-specific promoter has been utilized as a strategy to increase the replication potential of the virus in a tumor-specific manner. For example, promotors of cancer stem-cell-expressed genes such as Nestin and Musashi-1 have been exploited to drive ICP34.5 expression to increase virus replication in cancer cells, but not in normal adult cells where these promotors are not active [43,44,45]. rQnestin is one such tumor-promotor-driven oHSV that is currently being evaluated in patients with recurrent GBM (NCT03152318). While this is an area of excitement and development, readers are referred to a specific review on multiple approaches of using tumor-specific promotors to drive oncolytic viruses [46]. Gamma34.5 deletion in oHSV compromises virus replication. C130 and C134 genes from HCMV inhibit the PKR ability to arrest global translation by blocking eIF2α phosphorylation [47], but are not considered virulent. Chimera oHSV-C130 and oHSV-C134 have HCMV TRS1 gene (C130) or IRS1 (C134) inserted into the HSV-1 UL3/UL4 intergenic region. The chimera HSV significantly increases replication and spread in tumor [48]. Another chimera oHSV is oHSV–HF10, which has a natural deletion of UL43, UL49.5, UL55 and UL56 and insertions of UL53 and UL54 [49,50,51]. Preclinical and clinical studies show that oHSV–HF10 has high viral replication and cytopathic and bystander anti-tumor activity [52,53].

### 4.3. ICP4

Infected cell polypeptide 4 (ICP4) is an immediate early (IE) protein that is required for early and late gene transcription [54]. It contains a DNA-binding domain that enables it to bind to the viral genome, and interacts with the components of the cellular basal transcription apparatus to stimulate viral gene transcription. ICP4 is essential in the context of HSV growth and infection and its tumor-specific expression has been exploited to increase HSV replication in a tumor-specific manner. One innovative strategy exploited to deliver the tumor-specific expression of ICP4 has utilized tumor-specific miRNAs. miRNAs are small non-coding RNA that can bind and degrade cellular mRNA via the recognition of specific sequences in the 3”UTR of their target genes. miRNA-124 is an example of a miRNA that is lost in tumor cells but is highly expressed in neurons [55]. This property has been exploited to generate miRNA-driven oHSV wherein a miRNA-124 -recognition sequence inserted into the 3′ UTR of HSV ICP4 gene resulted in its expression in tumor cells that did not express the miRNA [56]. Thus, high levels of miRNA-124 in neurons lead to the recognition and disabling of ICP4 and ultimately block HSV replication. In contrast, the low-level expression of miRNA-124 allows for increased viral replication in glioma cells [56,57]. ICP4 expression under the control of a tissue-specific promoter has also been exploited to restrict efficient oHSV replication to tumor cells only [58,59]. For example, BM24-TE is an oHSV which exploits a Tcf binding site enhancer to drive ICP4 expression in Wnt/β-catenin-positive tumors such as colorectal cancer [58]. Similarly, ICP4 driven by the calponin promoter has also been shown to have enhanced specificity for calponin+ sarcomas [59].

### 4.4. ICP6

The UL39 gene in HSV-1 encodes for ICP6, a ribonucleotide reductase, which converts ribonucleotides to deoxy-ribonucleotides and is essential for HSV viral DNA replication. Quiescent and non-quiescent tumor cells with a dysregulation in the Rb tumor suppressor pathway upregulate the expression of the cellular ribonucleotidase and so create an environment that compensates for viral ICP6 loss. This permits for the efficient replication of a UL39-deleted HSV-1 in tumor cells. ICP6 also interacts with caspase 8 to block TNFα-mediated and Fas ligand-mediated apoptosis and necroptosis. Thus, ICP6 is mutated in several oHSVs, including the first generation oHSV (hrR3 [60]) and rRp450 [61]), second generation oHSV (G207) [62], and the third generation of oHSV (G47∆) [63].

### 4.5. ICP47

ICP47 is a viral protein that can downregulate antigen presentation activity by the inhibition of cellular transporter associated with antigen processing (TAP) protein. TAP protein is important for the delivery of antigenic peptides to nascent MHC class I molecules for presentation, a function designed to protect the virus from anti-virus immune recognition. Its deletion permits efficient antigen loading and thus can also activate anti-tumor immunity. Mutation of the ICP47 gene in oHSV thus increases immunogenicity and can augment the development of anti-tumor immune responses. ICP47 deletion also permits the early expression of US11, which can then counter cellular PKR-mediated anti-virus signaling and hence partially complement the loss of γ34.5. ICP47 mutation has been used in oHSV–G47∆ as a way to increase antigenicity and amplify virus-induced anti-cancer immunity [63].

### 4.6. Entry Tropism Modification

HSV-1 enters cells by endocytosis and also by fusion at the cell membrane via the specific interaction of viral glycoproteins with cell surface receptors. The virus glycoproteins responsible for fusion have been exploited to alter the tropism of oHSV. Tropism modification targets HSV membrane glycoproteins, such as glycoprotein D (gD) and glycoprotein B (gB), which mediate viral attachment and fusion into the cell membrane. HSV glycoproteins such as gD and gB bind to cell surface virus entry receptors such as HVEM and nectin-1 to facilitate viral entry. Nectin-1 and HVEM are ubiquitously expressed in human tissues and cells, thus mutants of gD and gB which are defective in binding to natural HSV-1 entry receptors are fused with ligands that show tumor-specific binding to permit for virus re-targeting. This approach is thought to increase tumor-specific virus entry while also reducing off-target toxicity. A recent study used this technology to target tumor-specific Her2, frequently overexpressed in many solid tumors including glioblastoma (GBM), breast cancer, and ovarian cancer. In this study, oHSV-LM249 and oHSV-LM113 were constructed via the insertion of scFV targeting HER2 into gD(amino acids 61-218) or gD (amino acids 6-38) [64,65]. The recombinant oHSV showed efficient targeting of Her2-positive tumor cells [51,66,67]. Targeting HSV gD has also been utilized to create EGFRvIII (oHSV-KNE), PSMA (oHSV R-593), EGFR (oHSV R611), and CEA (KNC-oHSV)-specific tumors such as glioma and HGG [68,69]. Besides gD, glycoprotein B (gB), another HSV transmembrane protein, allows for HSV binding, fusion and entry into the tumor cell. scFV-HER2 has been inserted into HSV gB (oHSV R909) and this demonstrates HER2+ tumor cell infection selectivity [70]. TAA-targeted oHSV: TAA is typically highly expressed in malignant tumors. Her2, EGFR and EGFRvIII are examples of TAAs expressed in many solid tumors. R-115 and R-LM113 oHSV are two distinct strains of oHSV that have been tested in GBM and HGG [67,71]. GBM can also be targeted by oHSV expressing EGFR or EGFRvIII, such as KNE, KGE [56,72].

## 5. Modulation of Cellular Signaling to Synergize with oHSV Therapy

Signaling transduction in the tumor cell or TME not only regulates tumor development and growth, but also controls the sensitivity to the treatments including chemotherapy, radiotherapy and immunotherapy (Figure 2). Drugs that can target these signaling pathways can both enhance and inhibit the effects of oHSV. Here, we will summarize some of the preclinical studies that have shown synergy between drugs that can inhibit these oncogenic pathways and oHSV therapy.

### 5.1. Proteasome Inhibitor

The host proteasome is important not only for the successful entry of oHSV into the host cells, but also for triggering anti-viral IFN responses in the infected macrophages via proteasome-mediated degradation of the viral capsid. Proteasome inhibitor induces estrogen receptor (ER) stress by the cellular aggregation of unfolded proteins, causing apoptosis in cancer cells and increased viral replication through the induction of heat shock protein 90 [73]. Despite the requirement of the cellular proteasome for virus replication, its inhibition leading to the induction of unfolded protein response (UPR) can also sensitize cancer cells to oncolysis [74]. Bortezomib is a proteasome inhibitor which exhibits a synergistic effect with oHSV. In fact, patients on long-term bortezomib are at risk for latent HSV reactivation [75]. The induction of UPR by bortezomib treatment improved therapeutic efficacy by enhancing oHSV replication and spread, and increased tumor cell lysis [76].

### 5.2. NOTCH

NOTCH signaling is frequently upregulated in numerous cancers, including glioblastoma (GBM), lung cancer, breast cancer and ovarian cancer (OC). Dysfunction of the NOTCH signaling pathway contributes to cancer cell proliferation, survival and angiogenesis. Furthermore, it maintains the stemness of cancer stem cells (CSCs) [77]. We recently described the activation of NOTCH signaling in oHSV-infected tumor cells. Infected cells could induce NOTCH activity in adjacent non-infected tumor cells. Mechanistically, we found that this was mediated through a virus-encoded miR-H16, which targeted and downregulated FIH1 (a direct target of miR-H16). FIH regulates MIB1-mediated NOTCH activation and thus its reduction induced NOTCH activity [78]. The combination of oHSV treatment with NOTCH inhibition by a gamma-secretase inhibitor (GSI) results in increased oHSV replication and spread among tumor cells and promotes survival compared with monotherapies [78]. The impact of this signaling on the tumor microenvironment remains to be explored.

### 5.3. HDAC

Histone deacetylase (HDAC) is an enzyme that inhibits oHSV infection and replication by inducing IFN regulatory genes in tumor cells. Additionally, aberrant expression and activity of HDAC promote tumor cell proliferation. Combining oHSV with HDAC inhibitors potentiates oHSV-induced anti-tumor immunity by impairing the innate anti-viral response, inhibiting IFN gene expression, leading to enhanced viral replication and spread [79,80].

### 5.4. cGAS-STING

cGAS senses dsDNA in virus-infected cells and activates STING, which phosphorylates and activates IRF3 to induce anti-viral type-I interferon signaling. The effect of STING signaling on oHSV replication appears to be context dependent, as it has been observed to have no effect on virus replication in hematologic malignancies, but a restrictive effect in melanoma [81,82]. C-GAS also senses damaged tumor cell DNA leaked into the cytosol after treatment with radiation or chemotherapy and can cooperate to elicit a strong anti-tumor immune response [83]. The combination of oHSV with cGAS-STING signal modification has been shown to increase the immunotherapeutic benefit of HSV virotherapy [84,85].

### 5.5. DNA Damage Response (DDR) Inducer

oHSV infection is known to induce a DNA damage response in tumor cells which in turn can enhance the replication of viral DNA. Chemotherapeutic agents target rapidly dividing tumor cells by inducing DNA damage. Temozolomide (TMZ), a chemotherapy drug for GBM, is a DNA-alkylating agent which can induce DNA damage. However, many GBM patients are resistant to TMZ treatment [86]. Combining oHSV-G47∆ with TMZ results in a significant increase in the infection and replication of oHSVs in GSCs and further increases the survival of mice bearing human GBM xenografts [87]. Radiation therapy is another strategy to induce DDR in tumor cells. Mezhir et al. showed that ionizing radiation induces oncolytic viral gene expression through p38 pathway and increases the anti-tumor efficacy [88]. Furthermore, the combination of oHSV with radiation imparts survival benefit and improves viral replication in recurrent GBM [89] and cholangiocarcinoma [90], respectively.

### 5.6. MEK Inhibitor

The mitogen-activated protein kinase (MAPK) signaling pathway is critical in regulating tumor progression via cell survival, proliferation and differentiation. The classical MAPK pathway consists of Ras, Raf, MEK and ERK, which relay cell growth and proliferation signals to the nucleus. BRAF, a proto-oncogene from the Raf family, is mutated in many solid tumors, including melanoma [91,92]. While hyperactivated MEK signaling in tumor cells was thought to be one of the pathways that was responsible for complementing the effect of ICP34.5-deleted viruses [93], surprisingly, recent studies have uncovered that the combination of their blockade with oHSV can have synergistic effects. Preclinical studies on MAPK inhibitors show a significant improvement in survival and sensitivity to immune checkpoint blockade [94]. Recent studies have shown that the combination of oncolytic virus T-VEC with trametinib, an MEK inhibitor, increased T-VEC replication not only in tumor cells in vitro, but also in tumor xenograft in vivo [95]. Furthermore, the combination of trametinib with oHSV also increases virus replication in immune “cold” tumor, glioblastoma, and induced tumor cell lysis [96].

### 5.7. PTEN/PI3K/AKT Signaling

The PI3K/AKT signaling pathway is a pro-mitotic, tumor-growth-progressing signaling pathway known to be activated in malignant cells. PTEN is a tumor suppressor that blocks this pathway and is frequently lost or bears loss-of-function mutations leading to cancer growth and progression. Inhibition of this pathway with small-molecule agents is thus a sought after area of cancer drug development. Interestingly, HSV-1 encodes for a viral gene, Us3, that mimics AKT kinase, and its deletion has been shown to synergize with phosphatidylinositol 3-kinase-Akt-targeting molecular therapeutics [97,98]. In these studies, PI3K inhibition in combination with an oHSV (deleted for viral US3) in GBM-bearing immune-deficient mice was shown to result in a significant benefit in anti-tumor efficacy. More recently, the restoration of PTEN via an oHSV was also shown to augment virus replication and also increase anti-tumor immunity [99].

## 6. Modulation of Tumor ECM to Enhance oHSV Therapy

### 6.1. Anti-Angiogenic Inhibitor

Angiogenesis is the development of vasculature and is increased during cancer progression. Blocking angiogenesis is thus a therapeutic strategy that aims to starve cancer cells [100]. Anti-angiogenic drugs such as bevacizumab and cilengitide affect vascular permeability and the inhibition of tumor growth [101,102]. An oHSV expressing anti-angiogenic vasculostatin which contains an integrin-antagonizing motif leads to the significant inhibition of glioma cell migration and invasion following bevacizumab treatment. This results in a significant extension of survival compared with bevacizumab monotherapy. The combination also increases oHSV infection and replication [103]. Another strategy demonstrated that chelating copper (Cu) could augment the efficacy of oHSV infection and replication by preventing angiogenesis and increasing the serum stability of oHSVs [104] (Figure 3).

### 6.2. Extracellular Immune Cells

In the tumor ECM, immune cells such as macrophages and NK cells are the major cellular players that orchestrate oHSV clearance. NK cells have been shown to be recruited to tumor microenvironment and contribute to the clearance of virus-infected cells [105,106]. Since NK cells can also have anti-tumor effects, the impact of NK cells on therapeutic efficacy has been a conundrum. Recent studies looking at the impact of NK cell numbers on viral efficacy have uncovered that the number of NK cells recruited to the TME is likely a critical determinant in evaluating their impact on therapy. Endogenous NK cells recruited to virus-treated tumors primarily target infected tumor cells, resulting in virus clearance and hence limiting therapeutic potential. However, when the NK activating signals after oHSV therapy are combined with a large number of exogenous CAR NK cells delivered to tumors, this activation can be harnessed to have an effective anti-cancer effect, resulting in improved therapy [76,107,108,109].

Myeloid cells such as macrophages have been shown to be recruited to the tumor microenvironment after oHSV therapy and are among the major producers of TNFα [110].

Blockade of TNFα produced by macrophages, using TNFα-blocking antibodies or the C16-mediated inhibition of STAT1/3 phosphorylation, can also increase oncolytic virus replication [111]. These macrophages also present a dual arm where they can clear oHSV but also assist in the development of antigen-specific T cell anti-tumor immunity. Additionally, recruited macrophages are educated by the tumor to develop into a tumor-supportive M2-like phenotype (TAM) that can block anti-tumor immunity. Consistent with these observations, the transient inhibition of TGF-β or macrophage depletion has been shown to augment virotherapy in immune-deficient mice [110,112,113,114]. It must be noted that while innate cellular responses can and do clear the oncolytic virus, their presence is also essential to activate anti-tumor immunity, an essential arm of oncolytic viral therapy. The white elephant issue in the field is the question of how antivirus innate immunity can be minimized while at the same time maximizing anti-tumor memory development. Future biomarker studies evaluating signatures predictive of response in patients on oncolytic virus therapy might help to clarify the dual relationship between innate and adaptive immune changes orchestrated in TME after virotherapy (Figure 3).

## 7. Strategies to Improve oHSV Delivery and Spread

### 7.1. Carrier Cells

The systemic delivery of oHSV has been challenging due to its rapid neutralization in serum. Complementarily, the mediated rapid neutralization of HSV-1 has been described, and its depletion has been shown to enhance oHSV propagation and efficiency [115]. Physiological levels of copper in the serum can also inhibit HSV-1, and the use of copper chelators such as ATN-224 has been shown to augment the serum half-life of oHSV therapy [104,116]. Anti-viral IgG and IgM also play a significant role in virus neutralization, and studies in rodent models have shown that its blockade by giving myelosuppressive doses of cyclophosphamide neutralizes viral infection [117,118]. Interestingly this approach is now being evaluated in combination with rQnestin34.5 in recurrent GBM patients (NCT03152318).

Some innovative “Trojan horse like” carrier cell approaches of delivering oHSV have been evaluated. One of the first approaches includes the use of mesenchymal stem cells (MSCs) as a carrier to deliver OVs for the treatment of GBM that can escape host cell immune surveillance [119]. Duebgen et al. demonstrated that MSCs loaded with oHSVs (MSC-oHSVs) can efficiently produce oHSV progeny and exhibit anti-tumor efficacy [120]. In a pre-clinical model of GBM resection, the intracranial implantation of artificial ECM-encapsulated MSC-oHSVs resulted in a significant increase in median survival compared to naked oHSVs. Furthermore, MSC loaded with oHSV–TRAIL (TNF-related apoptosis-inducing ligand) (MSC-oHSV-TRAIL) induced the apoptosis-mediated killing of GBM cells resistant to oHSV and TRAIL and efficiently prolonged the median survival of orthotopic tumor-bearing mice in immune-deficient GBM models. However, their efficacy against syngeneic GBM tumors is still unclear [121]. Another strategy to evade virus exposure to blocking antibodies is the use of syncytial strains of oHSV. Several preclinical studies using these viruses demonstrated potential therapeutic benefits [122,123], and future clinical trials will evaluate the safety and efficacy of this approach. Other cell types have also been employed as carrier cells to deliver oHSV, including neural progenitor cells [124] and antigen-specific T cells [125]. The utilization of neural progenitor cells to deliver oHSV not only prevents the clearance of virus by the immune system but the progenitor cells are able to differentiate into normal cells of the nervous system (astrocytes, neurons, and oligodendrocytes) to repair the brain destroyed by brain tumor [124]. oHSV loaded into T cells with low multiplicity of infection helps protect against antibody neutralization and allows for T cells to be delivered to the tumor environment [125] (Figure 4).

### 7.2. Increasing oHSV Spread into the Tumor Cells

An important prerequisite contributing to the efficacy of oncolytic virus therapy is viral infection of the tumor. Increased interstitial fluid pressure resulting from the permeable vasculature of the TME creates a physical barrier that limits the spread of viral vectors delivered intra-tumorally [126,127]. Consistent with this blocking treatment, induced edema with HMGB1-blocking antibodies or cilengitide has been shown to increase the therapeutic efficacy of this approach [102,128,129]. It must be noted that the reverse effect has been noted in situations where virus is delivered systemically [130,131,132]. Aside from interstitial pressure, extracellular tumor ECM has also been shown to limit the spread of oncolytic viruses. In a landmark study by Dr Jain’s group using multiphoton intravital fluorescent imaging combined with second harmonic generation signals, fibrillary collagen was observed to form a significant barrier that limited the spread of oHSV [126]. This led to the development of an oHSV encoding MMP9 MMP9/collagen [133,134]. Subsequent studies also discovered the importance of other tumor ECM constituents in forming a barrier that limited virus spread, and how these could be overcome by viruses that were trained to express enzymes that facilitated their digestion. For example, strategies have been developed to target tumor ECM hyaluronan [135], chondroitin sulfate [136], and collagen [137] to improve oncolytic virus efficacy, as a single agent and/or in combination with other therapeutics [138,139,140] (Figure 4).

Thus, while oHSV therapy is beginning to receive approval for treatment, the numerous approaches described here can be utilized to improve the impact of this treatment strategy. Future studies will help uncover the clinical significance of these approaches.

## 8. Future Directions

oHSV is a promising new therapeutic strategy for late-stage metastatic solid tumors that have not seen significantly effective treatments. The combination of oHSV with immune checkpoint inhibitors has been approved to help significantly increase the anti-tumor efficacy. Future investigations will revolve around identifying factors from oHSV and the TME which limit oHSV infection and replication. A reduction in those inhibitory factors will not only increase oHSV infection, replication, and tumor lysis activity, but also overcome immune suppression in the TME and induce a strong anti-tumor immune response.

## Figures and Tables

**Figure 1 viruses-14-00118-f001:**
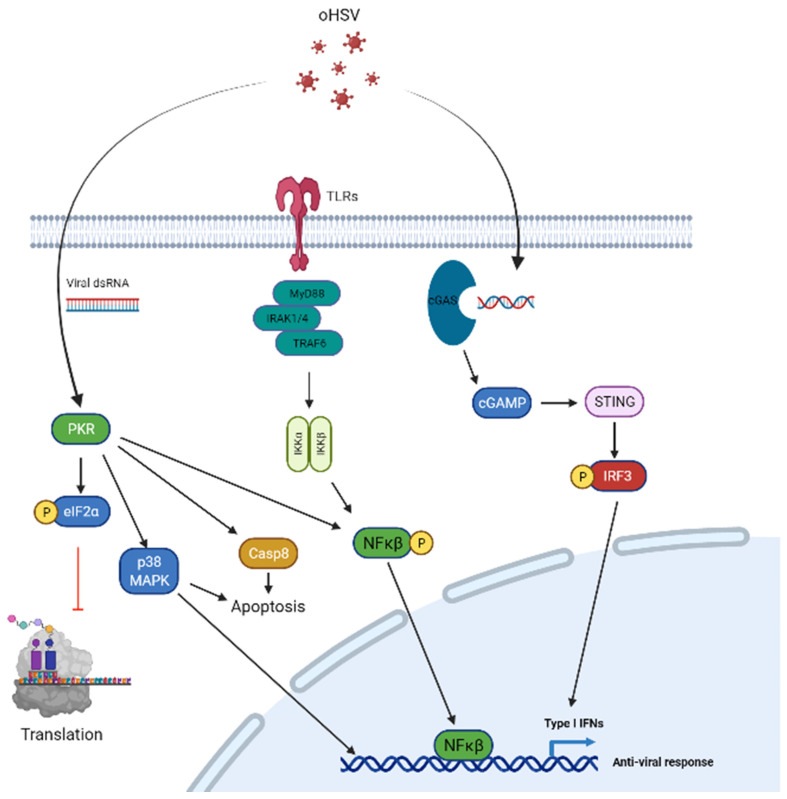
Signal transduction pathways in tumor cells and targeting for oHSV replication and spread. *PKR signaling* (**left**) confers a strong defense against viral infection which is activated by dsRNA from viral infection leading to auto-phosphorylation of PKR which phosphorylates eIF-2α. Phosphorylated eIF-2α leads to inhibition of cellular translation and prevents viral protein translation in the process. The PKR pathway is attenuated by oHSV via ICP34.5 which regulates PP1 and leads to PP1-mediated dephosphorylation of eIF-2α. VHS is an oHSV tegument nuclease that is able to degrade viral RNA and thus avoid PKR signaling activation. Furthermore, US11 of oHSV directly interacts with PKR and prevents its auto-phosphorylation. Additionally, PKR regulates cellular apoptosis pathways via Caspase 8 and p38-MAPK signaling, and also contributes to anti-viral response via p38–MAPK and NF-κB signaling pathways. *Toll-like receptors* (**middle**) are another pathway activated in infected cells and are able to detect viral nucleic acids. Upon entry of virus in the host cell, endosomes of infected cells detect viral nucleic acids leading to TLR activation. This leads to the recruitment of the adaptor protein, MyD88 and complexes with IRAK and TRAF6 ultimately leading to NK-κB activation and an anti-viral response. *cGAS-STING signaling* (**right**) is activated through viral dsDNA and is detected by cell cyclic GMP–AMP synthase(cGAS). Activation of cGAS leads to synthesis of 2′,3′-cGAMP and causes activation of its downstream target located on the cell’s endoplasmic reticulum, STING. Activation of STING leads to creation of autophagosomes and causes upregulation of Type I interferon gene expression and a strong anti-viral response. These pathways may be targeted with oHSV transgenes or innate features of oHSV to attenuate the host tumor cell anti-viral response.

**Figure 2 viruses-14-00118-f002:**
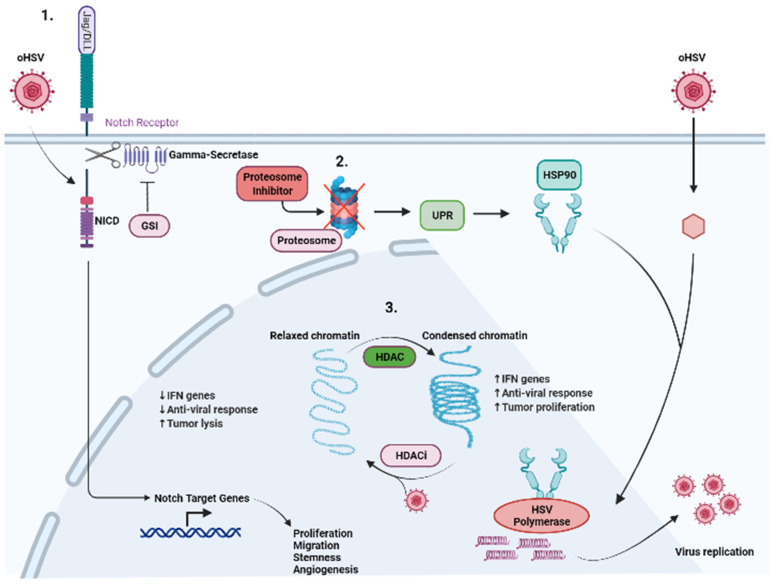
Modulation of cellular signaling to synergize with oHSV therapy. Multiple signaling pathways have been targeted in conjunction with oHSV therapy and highlighted are specific pathways which have been investigated. (1) Notch signaling highlights where Notch ligands from adjacent infected cells are binding to Notch receptors on uninfected cells. Binding of the Notch ligand causes a “pull” on the receptor, exposing the cleavage site on Notch-ICD (NICD). Gamma-secretase cleaves the NICD which subsequently leads to the expression of Notch target genes, leading to the growth, proliferation, and survival of tumor cells. Furthermore, oHSV infection can directly activate NOTCH signaling, leading to the cleavage and translocation of NICD to the nucleus, where it regulates tumor cell proliferation, stemness and angiogenesis. Gamma-secretase inhibitors in conjunction with oHSV have shown increased tumor cell killing and viral replication. (2) Cellular proteasomes are responsible for clearing viral proteins and viral capsids, and dampen the oHSV effect. Proteasome inhibition leads to an increase in UPR which induces HSP90 expression. HSP90 facilitates the translocation of HSV polymerase into the nucleus, leading to increased viral replication. The combination of oHSV and proteasome inhibitors leads to decreased IFN gene expression and anti-viral response as well as increased unfolded proteins and increased tumor cell killing. (3) Histone deacetylase (HDAC) is responsible for chromatin remodeling and changes relaxed chromatin into condensed chromatin. This leads to increased IFN gene expression and anti-viral immune response. Furthermore, this leads to increased tumor cell proliferation. The combination of HDAC inhibitors and oHSV has been shown to mitigate the effects of HDAC.

**Figure 3 viruses-14-00118-f003:**
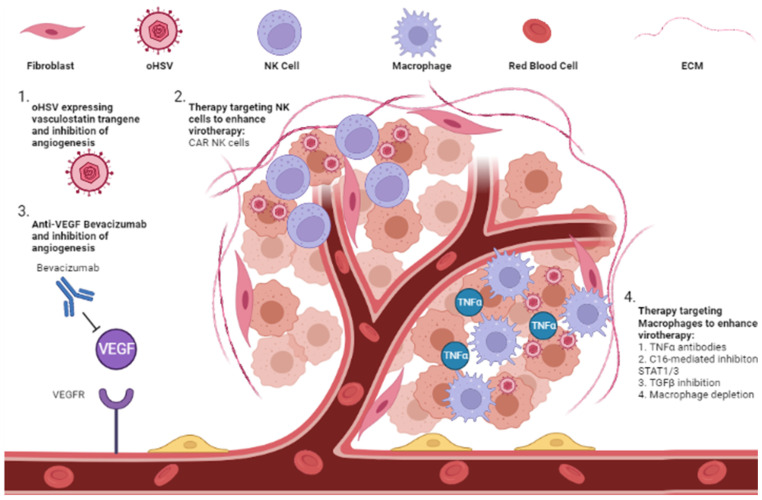
Modulation of tumor ECM to enhance oHSV therapy. Combination of oHSV with modulators of the ECM, including anti-angiogenesis therapies and therapies to modulate the anti-tumor response, have shown promising results. (1) oHSV expressing a vasculostatin transgene is able to attenuate tumor angiogenesis and synergistically lead to tumor cell killing. (2) NK cells are part of the ECM and are recruited to oHSV-infected tumor cells. NK cells are activated when tumor cells are infected, which leads to increased killing of infected cells and diminishes viral replication and spread. Exogenous CAR NK cells are a potential therapy to combine with oHSV to enhance viral replication and spread. (3) The anti-VEGF antibody, bevacizumab, leads to inhibition of VEGF binding with the VEGF receptor and leads to decreased angiogenesis and nutrient depletion of tumor cells. This has been combined with oHSV and led to increased tumor cell killing. (4) Macrophages also target oHSV-infected cells and secrete TNFα, leading to an anti-viral response preventing viral replication and spread. Antibodies targeting TNFα and C16-mediated blockade of STAT1/3 phosphorylation leads to increased viral efficacy through increased viral replication and spread. Macrophage depletion and TGFβ inhibition are further therapeutic strategies to enhance these effects.

**Figure 4 viruses-14-00118-f004:**
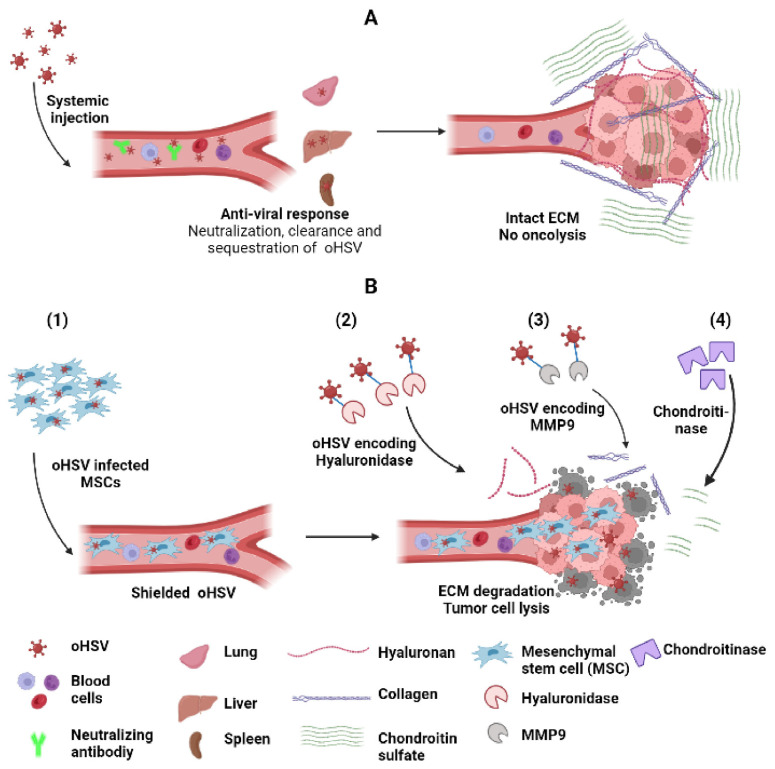
Strategies to improve systemic delivery and spread of oHSV. (**A**) Systemic delivery of oHSV to tumor cells is hampered by anti-viral response leading to neutralization by complement activation, interaction with blood cells and sequestration and clearance by macrophages in lung, liver (kuffer cells) and spleen. Intra-tumoral oHSV spread is limited by entrapment of oHSV in tumor ECM comprising collagen, hyaluronan and chondroitin sulfate. (**B**) Various strategies to improve oHSV systemic delivery and spread include (1) the shielding of oHSV by a suitable protective carrier such as MSCs, which facilitates escaping host immune surveillance, allows effective delivery to the tumor, and disrupts tumor ECM constituents, e.g., hyaluronan, collagen and chondroitin sulfate using (2) oHSV encoding hyaluronidase, (3) oHSV encoding MM9, and (4) ECM-degrading enzyme chondroitinase, respectively. These strategies eventually enhance oncolysis and augment oHSV-mediated antitumor effect.

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
