# Peer review of "Replication and Spread of Oncolytic Herpes Simplex Virus in Solid Tumors"

_viruses, 2022, doi:10.3390/v14010118_

Round 1
Reviewer 1 Report
The review by Hong et al focuses on the development of oncolytic HSV-1 for the treatment of solid tumors. The review does a reasonable job in highlighting the development of oHSV-1 and where improvements can be made. The manuscript though could be improved by addressing a number of issues.
- There a an apparent disconnect between Table 1 and what is actually discussed in the review. A greater alignment between the two would help with clarity. A footnote with abbreviations should also be included with Table 1.
- In the section “Infection and replication of HSV-1” should read “…the genome is repackaged and the resulting nucleocapsid fuses with the inner nuclear membrane. De-enveloped nucleocapsid then enters the cytoplasm….”
- In the “PKR signalling” section more information on HSV-1 encoded factors which regulate host PKR should be detailed. Are these also targets worth manipulating in context of oHSV-1?
- More information should be provided on approved HSV-1 TVEC (and use in combination with immune checkpoint therapy) and based on the modifications/strategies used in this design how can we improve i.e. use TVEC as starting point versus other “fresh” strategies as discussed in the review.
- In the section “Gamma 34.5” does ICP34.5 deletion really attenuate replication in tumor cells and neurons to the same degree? This should be clarified.
- Inclusion of at least one figure capturing strategies in oHSV design in relation to viral replication and host response would improve the review.
- Are there other examples of tumor-specific promoters in relation to gamma 34.5 expression?
- The authors need to make sure they have cited the most relevant recent reviews and papers in the field and include more examples of use of oHSV within each strategy. Currently the review is a bit sparse on detail.
- Under the section “Tropism modification…” HSV-1 not only enters by fusion at the cell membrane but also by endocytosis.
Reviewer 2 Report
In this review article the authors focus on oncolytic HSV replication and spread in solid tumors. Specifically, they highlight on the virus encoded genes and the pathways modulated by viral proteins to further improve virus infection and replication in tumor cells. The authors need the address the followings to further improve the MS.
Factors affect the spread of wild type HSV:
Apart from these innate sensing and anti-viral pathways, are there cancer cell specific pathways that can affect virus replication and spread? Adding those will help improving the MS.
Page 3, line 27: Use correct nomenclature for the TK deleted virus.
When describing the virulent genes that are modified, use additional references.
ICP4: need to add how is this connected to HSV OV.
Page 4, lines 32-33: elaborate how rQNestin-oHSV functions as OV.
In future, please add the line numbers for making review easier.
Page 5, lines 20-25: need sentence corrections.
Strategies to increase oHSV infection…………:
In this section very little is described for each of these pathways how they are targeted for enhanced oncolytic virotherapy. Immune regulation section is not clear.
Carrier cells: Only MSCs described. Other carrier cells?
A figure presenting the signal transduction pathways and how they are targeted for OHSV replication and spread will be informative.
Round 2
Reviewer 1 Report
The authors have improved the review significantly.
Minor comment: UL23 abbreviation for table 1 should indicate encodes "Thymidine kinase"
Reviewer 2 Report
none.